# Neurological Complications and Associated Risk Factors in Children Affected with Chronic Kidney Disease

**DOI:** 10.3390/children7060059

**Published:** 2020-06-08

**Authors:** Osama Safdar, Sulafa Sindi, Njood Nazer, Asmaa Milyani, Abdulrahman Makki

**Affiliations:** 1Pediatric Nephrology Center of Excellence, Faculty of Medicine, King Abdulaziz University, P.O.BOX 80215, Jeddah 21589, Saudi Arabia; 2Faculty of Medicine Department, King AbdulAziz University Hospital, P.O.BOX 80215, Jeddah 21589, Saudi Arabia; sulafah.sindi@gmail.com (S.S); nojoud.naz@gmail.com (N.N.); asmaa.milyani@gmail.com (A.M.); a.g.makki@gmail.com (A.M.)

**Keywords:** chronic, kidney, disease, neurological, complications, children

## Abstract

To investigate the correlation between chronic kidney disease (CKD) and the development of neurological disease among pediatric patients in Saudi Arabia. The present retrospective study recruited patients admitted to King Abdulaziz University Hospital during 2018. We reviewed electronic records to collect data on essential demographics including age, gender, and nationality; history of prior CNS disease or related symptoms; results of neurological physical examination; and findings of radiological investigations such as abdominal ultrasound, dimercaptosuccinic acid scan, micturating cystourethrogram, diethylene triamine pentaacetic acid scan, brain computed tomography, and magnetic resonance imaging. The most commonly diagnosed renal pathologies were neurogenic bladder and cystic kidney disease. The most common neurological manifestation was seizure disorder. Males were more frequently affected with neurological sequelae than females. The prevalence of neurological disorders was higher in patients over two years old. The most frequently observed stage of chronic kidney disease was stage 5. Most children who were affected with a neurological disorder required hemodialysis as part of their management plan. Patients with chronic kidney disease are at a high risk of neurocognitive defects. The type of management and renal diagnosis are significant factors that should be considered when anticipating central nervous system involvement in the case of chronic kidney disease.

## 1. Introduction

Kidney failure is a global health issue that is increasing both in incidence and prevalence, with poor resulting outcomes [1]. Data on the incidence and prevalence of chronic kidney disease (CKD) in the paediatric population worldwide is generally limited and is even more scarce in the Arab world [2]. Available data are from small studies in which extrapolated assumptions prove to be difficult. In 2005, one study from Kuwait estimated the incidence of CKD to be 28 per million of age-related population (pmarp) whereas the prevalence was as high as 329 pmarp [3]. Another study in 2002 from Jordan had estimated an incidence 11 cases of CKD pmarp with a prevalence of 51 pmarp [4]. Two studies from the western provinces of Saudi Arabia in 1997 and 2006 reported a similar mean incidence of 15.6 cases of CKD per million children and 9.2 cases of end-stage renal disease per million children [5,6]. Furthermore, it had outlined that the cause and course for CKD in Saudi Arabia is different from the reported etiology in Western countries [6].

Chronic kidney disease (CKD) is defined as either identified kidney pathology or glomerular filtration rate (GFR) of less than 60 mL/min/1.73 m^2^ for three months or more [1]. The condition is observed in 1–2% of the pediatric population with a gender bias, meaning that males are more frequently affected than females [7]. Common etiologies in the pediatric age group mainly involve congenital causes, such as anomalies of the kidney and urinary tract, and genetic disorders such as hereditary nephropathies and renal dysfunction as part of a syndromic disorder. However, acquired causes such glomerulonephritis are more likely to predominate in adolescent patients [8]. The impact of CKD extends to further organ systems including the central nervous system (CNS), with consequent effects on the development of intellectual and cognitive functions [9,10]. CNS involvement among children with advanced stages of CKD leads to increased morbidity and mortality [10]. A definitive association between CKD and CNS disease has been established, indicating the correlation of CKD with neurocognitive delays, especially in toddlers [11]. A study report of 13 pediatric patients with CKD since infancy reported that three (23%) of these patients had evidence of brain atrophy, contributing to the development of CNS complications including but nonconclusive to seizure disorder [12]. A cohort study of 33 pediatric patients with ESRD from congenital nephrotic syndrome were tested with brain MRI; it was documented that 19 (58%) of those patients had CNS infarcts or ischemia [12]. 

Developmental delay, hearing loss and encephalopathy are all complications that occur secondary to the onset of CKD [12]. The primary risk factors determining the severity of cognitive functions are duration and age at onset, as patients with earlier disease onset have poorer prognoses in terms of cognitive function [12]. Due to the paucity of available literature focusing on the association of CKD and CNS pathologies in Saudi Arabia, we aimed to investigate the correlation between CKD and the development of neurological disease among pediatric patients in Saudi Arabia.

## 2. Materials and Methods

### 2.1. Study Design and Recruitment

The present retrospective study involved the analysis of data obtained by review of electronic medical files of King Abdulaziz University Hospital. The study and its protocols were approved by the ethical committee at King Abdulaziz University Hospital prior to accessing patient files, and was registered at the national committee of biomedical ethics under HA-02-J-008 reference number 208-17. Furthermore, informed consent was obtained from all patients and their guardians. We used the Phoenix online medical records system to obtain the necessary data for patients admitted during 2018. A total of 1500 records were reviewed, of which only 425 patients qualified according to the following criteria: all children considered for the study should have been at an age ≤18 years and diagnosed with chronic kidney disease stage 3–5, which corresponds to an estimated glomerular filtration rate of <60 mL/min. The Schwartz formula was used for the calculation of estimated glomerular filtration rate (eGFR). Children were excluded from the study if they had a neurological diagnosis prior to the onset of chronic kidney disease or if their glomerular filtration rate was calculated at 60 mL/min or more. Few children in our sample had had complex disorders/syndromes as a cause for their developing chronic kidney disease, which may in part contribute to neurological complications. Such backgrounds include Alport syndrome, Williams syndrome, Fanconi Anaemia, sickle-cell anaemia, and systemic lupus erythematosus, and were among the renal diagnosis mentioned as “syndromic” or “other causes” in Table 1. Children suffering from neurological sequelae due to chronic kidney disease were regularly followed up in clinics in King Abdulaziz University Hospital. Clinical follow ups were updated in medical records with history, physical examination, work up as well as medications prescribed. Moreover, hospital admission and emergency visits were documented and reviewed during data collection.

### 2.2. Data Collection

We collected data on essential demographics such as age, gender, and nationality; history of CNS diagnosis or related symptoms; results of neurological physical examination; and findings of radiological investigations such as abdominal ultrasound, dimercaptosuccinic acid scan, micturating cystourethrogram, diethylene triamine pentaacetic acid scan, brain computed tomography, and magnetic resonance imaging. Patients suffering from any of the following were considered to be suffering from neurological disease: developmental delay, seizure disorder, neural tube defects, stroke, metabolic disease, cerebral palsy, or any other CNS diagnosis.

### 2.3. Statistical Analysis

All variables in this study were categorical and are therefore presented as frequencies and percentages. Pearson’s correlations test was performed to evaluate the correlation strength (*p*-values) between variables of interest. Missing data management was not required because all required data were collected correctly. The analysis was performed in 95% confidence interval using Statistical Package for Social Science (SPSS) software, version 23 (IBM, Armonk, NY, USA).

## 3. Results

A total of 425 cases were included in this study and among them, 276 (64.9%) were male and the remaining were female. More than three quarters (78.8%) of the cases were older than two years. The percentage of patients with CKD stage 3, stage 4 and stage 5 were 26.4%, 24.7% and 45.9%, respectively. Their specific renal diagnoses were given in Table 1. The clear majority (93.2%) had growth failure. Each renal diagnosis and specific CNS diagnoses are outlined in Table 1. CKD aetiology was acquired in 47%, congenital in 40%, and inherited in 13%. Consanguinity was found in 64% of the cases. The most common diagnosed renal pathologies were neurogenic bladder (16.9%), cystic kidney disease (15.3%), Nephrotic syndrome (12.9%), and posterior urethral valves (12.2%). The most common neurological manifestation was seizure disorder at a percentage of 14.1%. Additional neurological sequelae included developmental delay at 13.4% and stroke at 0.5%. Neurological associations included neural tube defects at 3.5% and cerebral palsy at 1.4%. Antiepileptic drugs were used by 7.3% of the cases. Most of the cases undergone CKD management conservatively (61.6%), while 17.4% needed hemodialysis, 12.2% needed peritoneal dialysis and the remaining 8.7% were treated surgically (Table 1).

Associations of different variables of interest were presented in Table 2. Odds ratios were generated for dichotomous variables. The statistically significant association between dichotomous variables were as follows: neurological disease and use of anti-epileptic drugs (OR 0.175, *p* < 0.001), renal diagnosis–syndromic causes and neurological disease (OR: 3.841, *p* = 0.011), tubular disease and growth failure (OR: 0.606, *p =* 0.032), syndromic causes and growth failure (OR: 0.112, *p* < 0.001), syndromic causes and developmental delay (OR: 0.189, *p* = 0.001), cystic disease and seizure disorder (OR: 3.888, *p* = 0.017) and HUS and seizure disorder (OR: 0.858, *p* = 0.034). Other associations are given in Table 2. 

Neurological disease was the most frequent in stage 5 CKD patients (39.5%) in comparison with stage 4 (33.3%) and stage 3 patients (23.2%). A higher percentage of male patients had neurological disease (35.1%) compared to female patients (28.9%). The percentage of neurological disease in cases older than three years was 34.3%, whereas it was lower among the patients who were younger than two years (27.8%). The highest prevalence of neurological disease was found in syndromic causes of renal pathology (64.3%) and in those who were managed with hemodialysis (47.3%) and peritoneal dialysis (40.3%). The percentage of neurological disease in each management and each renal diagnosis is given in Table 3. 

The frequencies and percentages of developmental delay, seizure disorder, neural tube disorder and CNS diagnosis for each renal diagnosis was presented in Table 4. The frequencies and percentages of each management in each age group were presented in Table 5.

## 4. Discussion

The National Kidney Foundation’s Kidney Disease Outcomes Quality Initiative classifies CKD into 5 stages. End-stage renal disease (ESRD) is defined as when kidney function has failed so significantly that the patient becomes dependent on dialysis with an eGFR below 15 mL/min/1.73 m^2^ [13].

Over the last 25 years, the association between ESRD and neurocognitive dysfunction has been established among pediatric patients, and its monumental contribution to the already increased morbidity and mortality of children with ESRD has been reported [12]. In our study, 33% of patients who were suffering from CKD stage 3–5 were diagnosed with some sort of neurological disorder. Of these, 39% were suffering from ESRD. This finding supports the relationship between the stage of CKD and incidence of neurological disease (*p* = 0.015). 

While some of the neurological implications stem from complications of renal failure, others may be a manifestation of an accompanying CNS pathology or neurological injury secondary to a different primary-organ pathology, such as hypoxia due to pulmonary hypoplasia which is often found in the case of obstructive uropathy [12]. This might explain our finding that, of all the underlying renal pathologies, only those associated with syndromic disorders were significant risk factors for neurological disease (*p* = 0.011). However, a high prevalence of neurological disease was still observed in other renal pathologies, such as obstructive uropathy (45%) and glomerulonephritis and tubular disease (42%). 

With regards to complications of renal failure, we found that the type of management plan (*p* = 0.009) and use of antiepileptic medications (*p* < 0.001) were the most significant risk factors. 

Previous studies during the 1970s and 1980s suggested that neurocognitive defects such as seizures were due to aluminum-induced neurotoxicity secondary to dialysis [14,15,16]. This proposal is supported by a study that compared the development outcomes of patients receiving conservative therapy to those who were dependent on dialysis. It was found that patients who remained on dialysis had worse development indexes than those receiving conservative therapy [17]. Additionally, another study on the outcomes of patients undergoing dialysis reported that 25% exhibited developmental delays [18]. In the present study, patients undergoing exclusive dialysis had the highest prevalence of neurological disease (44.4% of all patients who underwent either peritoneal or hemodialysis suffered from neurological disease). This is in contrast to the 28.6% of patients who underwent conservative therapy. Although dialysis is inevitable for patients with ESRD, our results highlight the need for careful evaluation of patients, and consideration of the balance between necessity for dialysis and the possible risk of adverse neurological outcomes.

One study showed that 3.7% of patients with CKD had experienced a convulsive disorder; consequently, we concluded that seizures are an occasional event that do not usually become chronic [19]. We found that specific renal pathologies were significantly associated with seizure disorders including cystic disease, obstruction, and hemolytic uremic syndrome, with respective *p*-values of 0.017, 0.036, and 0.034. (Figure 1).

Of the diagnoses associated with increased risk of developmental delay and growth failure (Figure 2), syndromic disorders were found to be associated with developmental delay (*p* < 0.001) and tubular disease with growth failure (*p* = 0.032). Our study found that age at diagnosis was not associated with developmental delay (*p* = 0.240). However, we did find that age at diagnosis significantly affected the course of the management plan (*p* < 0.001); 81.1% of patients diagnosed at ≤2 years were put on a conservative treatment plan, whereas 43.6% of those diagnosed at >2 years required either long-term dialysis or surgical intervention. This highlights the importance of considering factors that affect the management plan of patients with CKD in order to evaluate their risk of future neurological deficit. 

### 4.1. Limitations

This was a single-center retrospective study conducted at a tertiary institute. Results may be biased due to an increased sample of patients with advanced stages of CKD. Our study particularly focused on children with CKD stages 3 onwards, and therefore the incidence of neurological complications is expected to be higher than in those with early stage CKD. Furthermore, few children in our sample had had complex disorders/syndromes as a cause for their developing chronic kidney disease, which may in part contribute to further neurological complications. In this particular group, it is difficult to ascertain whether the neurological complications are exclusive to the CKD or may have been in part caused by the complex medical background. A higher prevalence of neurological disease was found in those who were managed with hemodialysis (47.3%) and peritoneal dialysis (40.3%). This could be in part related to this population being at a more critical stage of CKD as compared with those who were managed conservatively (28.6%). Again, there is no way to clinically ascertain with absolute confidence which of either is exclusively the cause of the neurological complication, and that is why our study focuses on the association of neurological disease in advanced stages of CKD and the possible factors that could indicate or alert to a more serious neurological outcome. 

### 4.2. Future Directions 

Further studies involving a multicentric prospective design and larger numbers of patients are required to confirm the conclusions of the present study.

## 5. Conclusions

Patients with chronic kidney disease are at a high risk of neurocognitive defects, of which some of the most prominent are developmental delay, growth failure and seizure disorder. The type of management and renal diagnosis are significant factors that are important considerations when anticipating central nervous system involvement.

## Figures and Tables

**Figure 1 children-07-00059-f001:**
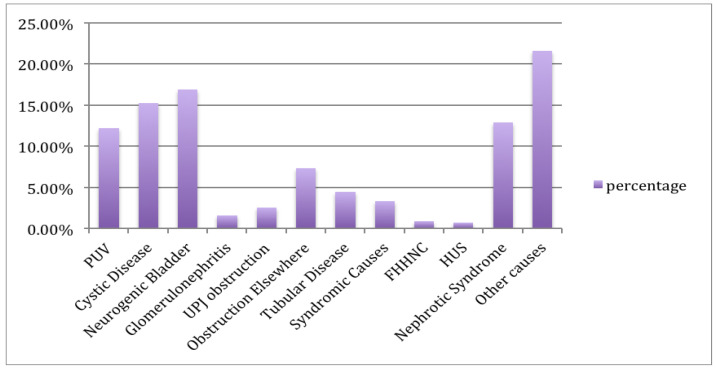
Frequencies of renal diagnoses. Abbreviations: FHHNC, Familial hypomagnesemia with hypercalciuria and nephrocalcinosis; HUS, hemolytic uremic syndrome; PUV, posterior urethral valves; UPJ, ureteropelvic junction.

**Figure 2 children-07-00059-f002:**
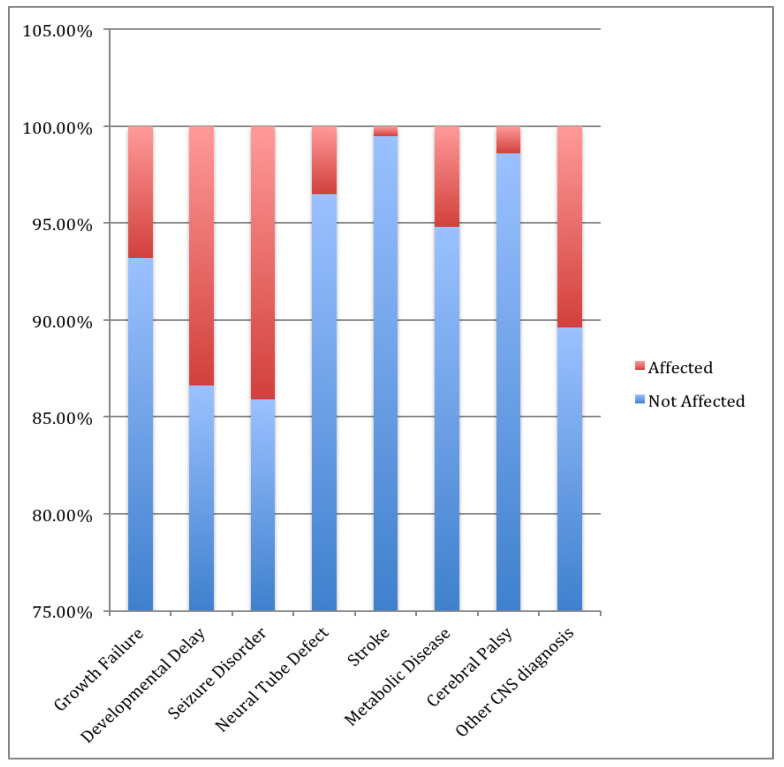
Frequencies of neurological diagnoses. Abbreviations: CNS, central nervous system. Red color reflects those who are affected with neurological disease. Blue color reflects those who are not affected with neurological disease.

**Table 1 children-07-00059-t001:** General descriptive studies of all variables (*n* = 425).

Characteristics	*N*	%
**Gender** Male Female	276 149	64.9 35.1
**Age** Younger than 2 years Older than 2 years	90 335	21.2 78.8
**Stage of CKD** Stage 3 (GFR 30–59) Stage 4 (GFR 15–29) Stage 5 (GFR < 15)	112 105 195	26.4 24.7 45.9
**Renal diagnosis** PUV Cystic disease Neurogenic Bladder Glomerulonephritis UPJ obstruction Obstruction otherwise Tubular disease Other causes Syndromic causes FHHNC HUS Nephrotic syndrome	52 65 72 7 11 31 19 92 14 4 3 55	12.2 15.3 16.9 1.6 2.6 7.3 4.5 21.6 3.3 0.9 0.7 12.9
**Growth Failure** Yes No	29 396	6.8 93.2
**Developmental Delay** Yes No	57 368	13.4 86.6
**Seizure Disorder** Yes No	60 365	14.1 85.9
**Neural Tube Defect** Yes No	15 410	3.5 96.5
**Stroke** Yes No	2 423	0.5 99.5
**Metabolic Disease** Yes No	22 403	5.2 94.8
**Cerebral Palsy** Yes No	6 419	1.4 98.6
**CNS diagnosis** None Tomography Muscle Tone disorder Hydrocephaly Encephalopathy Degenerative brain disease Pelizaeu Merzbacher disease	381 5 19 7 6 6 1	89.6 1.2 4.5 1.6 1.4 1.4 0.2
**Management** Conservative Peritoneal Dialysis Hemodialysis Surgical	262 52 74 37	61.6 12.2 17.4 8.7
**Antiepileptic Medication use** Yes No	31 394	7.3 92.7

Abbreviations: CKD, Chronic Kidney Disease; GFR, Glomerular Filtration Rate; PUV, Posterior Urethral Valve; UPJ, Uteropelvic Junction; FFHNC, Familial Hypomagnesaemia with Hypercalciuria and Nephrocalcinosis; HUS, Hemolytic Uremic Syndrome; CNS, Central Nervous System.

**Table 2 children-07-00059-t002:** Association between different variables by Chi-squared test (*n* = 425).

Correlation Between		OR (Yes/No)	*p*-Values
Variable 1	Variable 2		
Age of diagnosis	CKD Stage	-	**0.453**
	Growth failure	0.762	**0.591**
	Developmental delay	0.569	**0.156**
	Neurological disease	1.359	**0.240**
	Management	-	<**0.001**
Gender	CKD Stage	-	**0.946**
	Growth failure	0.875	**0.737**
	Developmental delay	1.092	**0.086**
	Neurological disease	0.749	**0.188**
Neurological disease	CKD stage	-	**0.015**
	Management	-	**0.009**
	Use of antiepileptic medication	0.175	<**0.001**
**Renal diagnosis** PUV Cystic disease Neurogenic Bladder Glomerulonephritis UPJ obstruction Obstruction otherwise Tubular disease Other causes Syndromic causes FHHNC HUS Nephrotic syndrome	Neurological disease	0.646 0.621 1.274 1.538 1.722 0.821 1.510 1.416 3.841 0.667 0.668 0.661	**0.193** **0.121** **0.366** **0.317** **0.371** **0.631** **0.385** **0.154** **0.011** **0.159** **0.223** **0.205**
**Each renal diagnosis** PUV Cystic disease Neurogenic Bladder Glomerulonephritis UPJ obstruction Obstruction otherwise Tubular disease Other causes Syndromic causes FHHNC HUS Nephrotic syndrome	Growth failure	0.862 1.609 1.296 0.931 0.725 1.067 0.606 1.351 0.112 0.931 0.931 1.310	**0.791** **0.443** **0.640** **0.470** **0.763** **0.932** **0.032** **0.551** <**0.001** **0.587** **0.683** **0.666**
**Renal diagnosis** PUV Cystic disease Neurogenic Bladder Glomerulonephritis UPJ obstruction Obstruction otherwise Tubular disease Other causes Syndromic causes FHHNC HUS Nephrotic syndrome	Developmental delay	2.765 1.623 0.952 0.928 1.564 1.049 0.818 0.667 0.189 0.865 0.865 1.071	**0.084** **0.282** **0.896** **0.945** **0.670** **0.931** **0.756** **0.206** **0.001** **0.429** **0.494** **0.873**
**Renal diagnosis** PUV Cystic disease Neurogenic Bladder Glomerulonephritis UPJ obstruction Obstruction otherwise Tubular disease Other causes Syndromic causes FHHNC HUS Nephrotic syndrome	Seizure disorder	1.298 3.888 0.621 0.986 0.426 1.579 0.439 0.802 0.394 0.857 0.858 1.396	**0.569** **0.017** **0.154** **0.990** **0.062** **0.036** **0.076** **0.496** **0.077** **0.415** **0.034** **0.464**

**Table 3 children-07-00059-t003:** Frequency of neurological disease in each CKD stage, management, use of antiepileptic medications, renal diagnosis gender and age groups.

Different Groups of Interest	Neurological Disease	
	*N*	%
**CKD stage** Stage 3 (*n* = 112) Stage 4 (*n* = 105) Stage 5 (*n* = 195)	26 35 77	23.2 33.3 39.5
**Management** Conservative (*n* = 262) Peritoneal dialysis (*n* = 52) Hemodialysis (*n* = 74) Surgical (*n* = 37)	75 21 35 9	28.6 40.4 47.3 24.3
**Use of antiepileptic medications** Yes (*n* = 31) No (*n* = 394)	22 118	71.0 29.9
**Renal diagnosis** PUV (*n* = 52) Cystic disease (*n* = 65) Neurogenic Bladder (*n* = 72) Glomerulonephritis (*n* = 7) UPJ obstruction (*n* = 11) Obstruction otherwise (*n* = 31) Tubular disease (*n* = 19) Other causes (*n* = 92) Syndromic causes (*n* = 14) FHHNC (*n* = 4) HUS (*n* = 3) Nephrotic syndrome (*n* = 55)	13 16 27 3 5 9 8 36 9 0 0 14	25.0 24.6 37.5 42.9 45.5 29.0 42.1 39.1 64.3 0.0 0.0 25.5
**Gender** Male (*n* = 276) Female (*n* = 149)	97 43	35.1 28.9
**Age group** Younger than 2 years (*n* = 90) Older than 2 years (*n* = 335)	25 115	27.8 34.3

**Table 4 children-07-00059-t004:** Frequency of developmental delay, seizure disorder, NTD (Neural Tube Defects) and CNS (Central Nervous System) diagnosis in each renal diagnosis.

Renal Diagnosis	Developmental Delay *N* (%)	Seizure Disorder *N* (%)	NTD *N* (%)	CNS Diagnosis *N* (%)
PUV (*n* = 52) Cystic disease (*n* = 65) Neurogenic Bladder (*n* = 72) Glomerulonephritis (*n* = 7) UPJ obstruction (*n* = 11) Obstruction otherwise (*n* = 31) Tubular disease (*n* = 19) Other causes (*n* = 92) Syndromic causes (*n* = 14) FHHNC (*n* = 4) HUS (*n* = 3) Nephrotic syndrome (*n* = 55)	3 (5.8) 6 (9.2) 10 (13.9) 1 (14.3) 1 (9.1) 4 (12.9) 3 (15.8) 16 (17.4) 6 (42.9) 0 (0.0) 0 (0.0) 7 (12.7)	6 (11.5) 3 (4.6) 14 (19.4) 1 (14.3) 3 (27.3) 3 (9.7) 5 (26.3) 15 (16.3) 4 (28.6) 0 (0.0) 0 (0.0) 6 (10.9)	2 (3.8) 3 (4.6) 1 (1.4) 0 (0.0) 1 (9.1) 0 (0.0) 0 (0.0) 7 (7.6) 1 (7.1) 0 (0.0) 0 (0.0) 0 (0.0)	3 (5.8) 2 (3.0) 13 (18.1) 2 (28.6) 2 (18.2) 4 (13.0) 1 (5.3) 10 (11.0) 4 (28.5) 0 (0.0) 0 (0.0) 3 (5.4)

**Table 5 children-07-00059-t005:** Frequency of each management in each age group.

Age Group	Conservative *N* (%)	Peritoneal Dialysis *N* (%)	Hemodialysis *N* (%)	Surgical *N* (%)
Younger than 2 years (*n* = 90)	73 (81.1)	7 (7.8)	3 (3.3)	7 (7.8)
Older than 2 years (*n* = 335)	189 (56.4)	45 (13.4)	71 (21.2)	30 (9.0)

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
