# Peer review of "Neurological Complications and Associated Risk Factors in Children Affected with Chronic Kidney Disease"

_children, 2020, doi:10.3390/children7060059_

Round 1

Reviewer 1 Report

Safder et al report on the neurological complications of chronic kidney disease in Saudi Arabia. This is a single institution retrospective study.

THe population is quite different than that studied in most other studie. The common pathologies were neurogenic bladder and cystic kidney disease. It is not surprizing that neurogenic bladder is associated with neurological disease though not always clear that a consequence or common pathology.

Majority of patients with neurological complications required hemodialysis so represented end stage renal disease. This population is known to have much higher comorbidiyy and complications and this needs to be acknowledged.

The authors should specifically comment on proportion of cases with renal disease due to genetic causes as these often have a high rate of associated abnormaities

Reviewer 2 Report

Dear Author

your manuscript has been carefully evaluated. I think that your manuscript could be interesting but needs some improvements.

In specific:

- In the introduction, in the first two lines, I would add data relating to the incidence and prevalence of renal failure in children with the related bibliographical data (if we speak in a generic way then we will have to enter data relating to acute and chronic form otherwise it is necessary to specify which condition referenced).

- In the materials and methods section it is better to describe the population under study specifying the nature of the renal failure from which the affected patients (some data is provided in other parts of the manuscript). In fact, it is necessary to specify in which cases renal failure is part of more complex pathologies in which the nervous system is involved. In these cases, it is not possible to discuss neurological complications that have occurred in patients with renal insufficiency

- In the results section there are references to the tables but I believe that the main data must still be expressed in the text, so as to give immediate and clear information to the reader

- The authors do not refer to the way in which patients with renal insufficiency have to undergo periodic neurological evaluations: follow-up methods and times for these patients? Gods or a proposal could improve clinical practice

- Implement data and bibliography on dialysis-damage correlation to the nervous system and causes of seizure in patients with kidney failure

Round 2

Reviewer 2 Report

Dear Author,
I carefully evaluated the manuscript to which the suggested changes were made.
The study thus prepared appears interesting, correctly presented. The data introduced in the drafting of the manuscript make it more complete and clear, provide complete and useful information that improve its scientific value